# Application of ANN for Analysis of Hole Accuracy and Drilling Temperature When Drilling CFRP/Ti Alloy Stacks

**DOI:** 10.3390/ma15051940

**Published:** 2022-03-05

**Authors:** Vitalii Kolesnyk, Jozef Peterka, Oleksandr Alekseev, Anna Neshta, Jinyang Xu, Bohdan Lysenko, Martin Sahul, Jozef Martinovič, Jakub Hrbal

**Affiliations:** 1Department of Manufacturing Engineering, Machines and Tools, Sumy State University, Rymskogo-Korsakova Str., 2, 40007 Sumy, Ukraine; v.kolesnik@tmvi.sumdu.edu.ua (V.K.); o.alekseev@tmvi.sumdu.edu.ua (O.A.); anna_neshta@tmvi.sumdu.edu.ua (A.N.); b.lysenko@tmvi.sumdu.edu.ua (B.L.); 2Faculty of Materials Science and Technology, Slovak University of Technology in Bratislava, Ulica Jána Bottu č. 2781/25, 917-23 Trnava, Slovakia; martin.sahul@stuba.sk (M.S.); jozef.martinovic@stuba.sk (J.M.); jakub.hrbal@stuba.sk (J.H.); 3School of Mechanical Engineering, Shanghai Jiao Tong University, Shanghai 200240, China; xujinyang@sjtu.edu.cn

**Keywords:** CFRP/Ti alloy stacks, thermocouple method, drilling temperatures, hole diameter, out of roundness, ANN analysis

## Abstract

Drilling of Carbon Fiber-Reinforced Plastic/Titanium alloy (CFRP/Ti) stacks represents one of the most widely used machining methods for making holes to fasten assemblies in civil aircraft. However, poor machinability of CFRP/Ti stacks in combination with the inhomogeneous behavior of CFRP and Ti alloy face manufacturing and scientific community with a problem of defining significant factors and conditions for ensuring hole quality in the CFRP/Ti alloy stacks. Herein, we investigate the effects of drilling parameters on drilling temperature and hole quality in CFRP/Ti alloy stacks by applying an artificial neuron network (ANN). We varied cutting speed, feed rate, and time delay factors according to the factorial design L_9_ Taguchi orthogonal array and measured the drilling temperature, hole diameter, and out of roundness by using a thermocouple and coordinate measuring machine methods for ANN analysis. The results show that the drilling temperature was sensitive to the effect of stack material layer, cutting speed, and time delay factors. The hole diameter was mainly affected by feed, stack material layer, and time delay, while out of roundness was influenced by the time delay, stack material layer, and cutting speed. Overall, ANN can be used for the identification of the drilling parameters–hole quality relationship.

## 1. Introduction

Carbon fiber reinforced plastics (CFRP) are widely used in different applications, starting with rope threads [1], gears [2], and aerospace components [3]. However, the poor machinability of CFRP complicates the manufacturing process of final parts. In aerospace components, CFRP is usually connected with titanium and aluminum alloys to form a hybrid stack [3]. Drilling is the most widely used machining operation when producing holes for clamping FRP/metal stacks in the aerospace industry [4,5]. To ensure the proper service life of joints, the drilled holes in FRP/metal stacks must satisfy certain quality parameters [6,7]. These quality parameters are hole diameter [8,9,10,11], roundness and cylindricity [12,13], surface roughness [14,15,16], delamination factor (*F_d_*) [8,14,17,18], thermal destruction [19] and damage value (*Q_d_*) [20], and burr height for Ti alloy [21,22,23]. Milentiev et al. reported the connection between tool geometry and process parameters on delamination factor in CFRP [24]. Overall, drilling of FRP/metal stacks is the critical machining operation in the manufacturing chain of aircraft components, which must be precisely controlled to achieve the required hole quality parameters.

Various factors were reported to affect the hole quality parameters in CFRP/Ti alloy stacks, including drill bit wear rate [6,25,26,27,28,29], cutting tool coating [4,15,22,26,30,31,32], cutting parameters [10,11,14,33,34], drill bit geometry [8,21,35,36], dynamic characteristics of CFRP [37], drilling strategies, and techniques, namely one-shot drilling of the stack [10,11,14,22,33,35,38,39,40,41], stepped bit geometry drilling [8,21], pilot hole drilling [42], cryogenic drilling [43,44], minimal quantity lubrication (MQL) [29,45], helical milling [9,46], and drilling with core drill [47]. In such a way, it can be summarized that one-shot drilling of CFRP/Ti alloy stacks is still a mainstream technology of manufacturing holes studies; we concluded that the most promising factors to improve the hole quality are the cutting parameters of the one-shot drilling strategy. The main parameters, namely cutting speed and feed rate, are typically varied in the range from 10 m/min to 150 m/min and from 0.025 mm/rev to 0.1 mm/rev, respectively [3]. The good drill bit geometry for one-shot drilling was defined as drill point angle (135°–140°) [11,48,49], helix angle (22°–35°) [8,42], and clearance angle (5°–10°) [13]. Shu et al. reported about the reduction in drilling temperature in CFRP from 160 °C to 90 °C, which corresponded to conventional and dedicated drill bit geometry with a double margin [50]. At present, it is hard to make certain conclusions concerning the proper coating of the drill bit when drilling stacks because researchers give quite controversial conclusions concerning the effectiveness of this or that coating [51]. The helical milling technique for hole production can be considered a perspective direction of feature research [52]. However, it has a disadvantage of comparatively big machining time, limiting the broad implementation of this technique in production [9,53]. MQL technique offers to reduce tool wear by up to 20% [54]. However, the complexity of MQL supply to the cutting zone still is the main limiting factor for its usage in production conditions [45]. The cryogenic drilling technique does not prove its effectiveness in ensuring hole quality in CFRP/Ti alloy stacks. Still, the low temperatures’ effect leads to decreasing CFRP fatigue characteristics. In such a way, it can be summarized that one-shot drilling of CFRP/Ti alloy stacks is still a mainstream technology of manufacturing holes.

The physics of one-shot drilling strategy when making holes in CFRP/Ti alloy stacks was studied in terms of the generated torque [21,33], thrust force [4,8,41,55,56], frictional heat [57], acoustic emission [58], wear rate [27,38,59], chip formation [13,22,43,60], and drilling temperature [11,19,40,55,56]. The last factor was reported as crucial for hole quality assurance. The drilling temperature was measured by embedding a thermocouple [10,11,40] and optical fiber [38] in the drill bit and the workpiece [19,55,56]. The temperature distribution was also studied via the thermographic method [56,61,62] finite element analysis for drill bit [11] and CFRP/Ti alloy stack [13]. It was outlined that at present, the drilling temperature is the critical physical factor affecting the hole quality, while the thermocouple is one of the most reliable methods of studying it when machining FRP/Metal stacks [40]. The scientific conclusions of experimental studies about the significance of the influence of various factors are based on the mathematical analysis of experimental data.

The most common mathematical tools for analyzing experimental data are the analysis of variances ANOVA [63,64,65]. However, the traditional ANOVA method does not provide reliable information on many factors affecting the hole quality parameters when drilling CFRP/Ti alloy stacks. From this point of view, ANN analysis can provide a researcher with reliable calculations and predictions of possible input–output connections. Temporary trends of experimental data analysis in engineering are utilized by artificial neuron networks (ANN), fuzzy logic, Autoregressive Integrated Moving Average (ARIMA) [66], and machine-learning methods [67]. ANN proves its applicability as a universal tool for analysis of various not only in non-technical [68] but also in technical processes such as: hydro-mechanical [69], cutting force when grinding [70], prediction of milling process parameters [71], surface roughness when turning [72], and milling [73]. It was suggested to use ANN to analyze and optimize cutting parameters when turning Ti-6Al-4V, predicting surface roughness and cutting force [74], and material removal rate [75]. The most typical ANN architectures for analysis and prediction of the above-mentioned output machining process parameters are back-propagation neuron network (BPNN) [74,76,77], conventional neuron network [78], and multi-layer perceptron (MLP) [72,75,79]. The ANN analysis is ensured by the Levenberg–Marquardt backpropagation algorithm [72,73,76,77,79] and scaled conjugate gradient training algorithm [75], particle swarm optimization algorithm [80], and Bayesian regularization [74]. The application of ANN for analysis of the experimental results of CFRP drilling was dedicated to predicting tool wear based on thrust force analysis [25,81]. Han et al. reported the applicability of the discrete wavelet transformation technique to predict tool wear with a maximum estimation error of about 22% and the average error ≤ 8%. The ANN with cascade-forward back propagation architecture was utilized for smart decision making of the tool life, based on the analysis of thurst force and torque into frequency domain [82]. Cui et al., provided long short-term memory (LSTM), temporal convolutional network (TCN) and singular spectrum analysis (SSA) for prediction of CFRP delamination when drilling taking in consideration results of experimental data of thrust force, drilling temperature and vibrations [83]. Implementation of SSA lead to 39% reduction in delamination prediction error in comparation to traditional regression method.

The introduction outlined that single-shot drilling is still a leading technique of drilling holes in CFRP/Ti alloy stacks. However, some alternative techniques such as ultrasonic drilling, vibration-assisted drilling, helical milling, cryogenic, and drilling with MQL are the focus of research activities. The most significant factors affecting hole quality in CFRP/Ti alloy stacks are cutting parameters, drill bit geometry, tool coating. At the same time, the physical basis of the cutting process when drilling is studied via measurement of thrust force, torque, chip formation, wear rate, drilling temperature, and vibrations. Though many studies were conducted for solving the engineering problem of ensuring hole quality parameters in CFRP/Ti alloy stacks, namely: hole diameter, roundness and cylindricity, roughness, delamination factor, damage factor, fiber pull out, uncut fiber, and burr size. Ensuring hole quality in CFRP/Ti alloy stacks cannot be considered solved. Considering that during the last decade, many experimental results were accumulated the machine learning methods, ANN and fuzzy logic became mainstream methods that displace traditional ANOVA methods for experimental data analysis, prediction of machining output parameters, and feature decision-making strategies. Nowadays, ANN is used for the prediction of surface roughness, tool wear, and cutting force. In means of application, ANN for drilling CFRP or CFRP/Ti alloy stack analysis parameters were utilized for delamination, tool wear, and thrust force prediction. However, drilling temperature, hole diameters, and out of roundness have not yet been analyzed via ANN when drilling CFRP/Ti alloy stacks.

Based on the above literature review, it can be concluded that the use of ANN analysis of experimental data will be a suitable opportunity to fill the research gap of research on the subject. This provides ANN analysis of process parameters influence on drilling temperature, hole diameters, and out of roundness when drilling CFRP/Ti alloy stacks. Such analysis presented novel results of technological factors’ effect on hole quality parameters in CFRP/Ti alloys stacks considering the influence of time delay factor, which was not reported earlier. The experimental data were gained via an experimental study constructed based on the design of the experiment according to the Taguchi method consisted of nine tests in which three factors (cutting speed, feed, and time delay) varied on three levels. The drilling temperature was measured in real-time in the drill bit by a K-type thermocouple via a wireless device. The hole diameters and out of roundness were measured via the coordinate measuring machine method. The experimental data featuring ANN analysis were provided via multilayer perceptron (MLP) and radial basis function (RBF). The results demonstrate that ANN can be an effective tool for the identification of the drilling parameters–hole quality and drilling temperature relationship.

## 2. Materials and Methods

### 2.1. Workpiece Material and Cutting Tool

A two-layer adhesive stack, consisting of CFRP and titanium alloy plates 95 × 200 mm, were used in experiments. The CFRP laminate contained 45 unidirectional plies of 0.20 mm thickness, made of IM7 carbon fiber, and Larit (LR285) epoxy resin with the following stacking sequence [0^2^/90^2^] fabricated by hand lay-up technique, vacuum bag molding using vacuum pump in a controlled atmosphere. The total thickness of the CFRP plate is 9 ± 0.01 mm, with 60% fiber volume content [11]. The material of titanium alloy plate was chosen Ti-2.5Al-2Mn near α, alloy with a thickness of 8 mm, with the following mechanical properties: tensile strength of 735 MPa, elasticity modulus of 115 GPa, the density of 4550 kg/m^3^, and hardness of 178 HV [84]. In such a way, the total thickness of the CFRP/Ti alloy stack was 17 mm. The actual chemical composition of Ti alloy was defined at JEOL (JOEL Ltd., Tokyo, Japan) JSM-7600F scanning electron microscope (SEM) via dispersion spectroscopy (EDS) (Table 1).

For the experimental study, nine WC9 TiN-TiAlN coated Ø10 mm twist drill 5510-R-RT100U Guhring (Gühring Slovakia s.r.o., Považská Bystrica, Slovakia) items were used. The actual geometry of each drill bit was measured at a universal automatic measuring machine for cutting tool Zoller Genius 3s (E. ZOLLER GmbH and Co., KG, Pleidelsheim, Germany) (Table 2).

### 2.2. Proposed Approach for Drilling Temperature, Hole Diameter, and Roundness Prediction

The experimental data were gathered and proceeded offline and online procedures (Figure 1). The thrust force was measured using the piezoelectrical method three-component dynamometer Kistler 9257. The signal of thrust force was transmitted to a multi-channel amplifier, Kistler 5070 (Kistler Group, Winterthur, Switzerland). Measurement of drilling temperature was utilized by wireless measurement device (WICUTEM). Raw experimental data of thrust force and drilling temperature signals were synchronized with a computer (astronomic) time criterion at a personal computer to identify machining process start. The data for the hole diameter and out of roundness were collected at CNC coordinate measurement machine ZEISS PRISMO ULTRA (Carl Zeiss AG, Oberkochen, Germany). The experimental data set was combined in a single data file that contained drilling temperature, hole diameter, and out of roundness in respect to cutting parameters, which were evaluated along with the depth of the machined hole. The creation made synchronization of drilling temperature, hole diameter, and out of roundness of depth of the hole, which was calculated as a function of machining time, cutting speed, feed, and time delay.

### 2.3. Experimental Set-Up

The present research’s experimental set-up was assembled at 5 axial DMU 85V CNC (Computer Numerical Control) milling center (DMG MORI, Pfronten, Germany) (Figure 2). The 5-axis DMU 85 V is a 5-axis machining center suitable for the production of complex components in various areas such as healthcare, energy, aerospace, and the automotive industry. The machining center offers simultaneous 5-axis high-speed milling with high dynamics and high cutting power torque. The machine was equipped with stiff construction with high balanced moving parts and static masses. The spindle range was up to 30,000 rpm or up to 430 Nm. The maximal workpiece diameter is 1040 mm.

A fixture of CFRP/Ti alloy sample was made in precise wise Schunk Kontec KSC-F-125 (SCHUNK GmbH and Co. KG, Lauffen/Neckar, Germany) [85] with the error of location 0.02 mm. Thrust force was measured with a three-component dynamometer Kistler 9257 (Kistler Group, Winterthur, Switzerland) sampling frequency, 200 Hz. Drilling temperature was measured via a wireless temperature measuring device (WICUTEM), fixed at HSK40 collet chuck with collet screw, and rotated with CNC machine spindle. The online measurement of drilling temperature was realized by combining the thermocouple method with the wireless high-frequency transmission of the signal via Bluetooth channel. Feature data collection at a personal computer (PC) were implemented in a self-designed WICUTEM software interface. The OMEGA (Omega Engineering Inc., Norwalk, CT, USA) (XCIB-K-2-5-3) K-type thermocouple sensor with the operational range from 0 °C to 1040 °C was embedded under the flank surface of the WC twist drill (Figure 2a). The uncertainty of measurement was defined as ±1.1 °C. The contact between the thermocouple and WC twist drill surface was ensured by high-temperature assembly compound GRIPCOTT NF (MOLYDAL SA, Saint-Maximin Cedex, France) with an operating temperature range of 25 °C~1000 °C. The sampling frequency of the WICUTEM device was 200 Hz, and the signal transmission frequency was 2.4 GHz.

The drilling of the CFRP/Ti alloy stack was realized according to factorial design L_9_ (34) Taguchi orthogonal array (Table 3 and Table 4). In each test, 23 holes were drilled. The drilling procedure was provided under the single-shot technique with the same process parameters for both CFRP and Ti alloy layers of the stack. The variable factors were: cutting speed varied on three levels from 15 m/min to 65 m/min, and the feed rate varied, 0.02 mm/r–0.08 mm/r. In order to approach the real production conditions, we also varied the drill bit temperature from the so-called “cold drill machining” (CDM) to “hot drill machining” (HDM) conditions. Under CDM conditions, the drill bit was cooled to a room temperature (20 ± 2 °C) for drilling each hole, which in the time domain corresponded to 120 s time delay (*T_d_*) prior to drilling the next hole (Figure 2b). Drilling under HDM conditions corresponded to the time delay (*T_d_*) from 5 s to 10 s between the adjacent holes (Figure 2c).

The hole diameter and out of roundness were measured at coordinate measuring machine ZEISS PRISMO ULTRA (Carl Zeiss AG, Oberkochen, Germany). The diameter of the probe was 3 mm. The accuracy of measurement was within 0.1 µm. The actual measurement results were rounded to 1 µm. The measuring procedure was implemented according to the measuring scheme (Figure 3). The hole diameters and out of roundness factor were assessed based on the 24 cross-sections of CFRP/Ti alloy stack, each taken 0.5 mm along the hole depth from 0.5 mm to 15 mm for each hole. The hole profile on the cross-sections was mapped with 32 points to quantify the values of the diameter and roundness. In such a way, the cloud of 768 data points was obtained for each hole. The measured outputs of the hole accuracy parameters were later considered in the ANN testing model.

### 2.4. Methodology of ANN Analysis

The mathematical apparatus of neural networks [86,87] was used to analyze quantitative data obtained in the experimental study. For this purpose, the neural networks with direct signal distribution were examined: the Multilayer Perceptron (MLP) and the Radial Basis Functions (RBF) networks. The architecture of such networks is built on a multilayer structure. Based on the literature review, the MLP/RBF ANN architecture was selected as one of the most commonly used ANN architectures, which proves its reliability in analyzing various experimental data in engineering. It includes interconnected neurons of the input layer, one or several intermediate (hidden) layers, and the output layer (Figure 4).

A supervised learning method was used to move from the general architecture of neural networks to the network configured to find solutions specific to tasks of analysis and predict output factors for a drilling operation in the CFRP/Ti alloy stack. For this, the neural networks with different training parameters were trained based on a data set obtained experimentally, and then a network with the most preferred characteristics was selected.

The measured numerical values of the parameters of the input factors and output response of the drilling operation were presented as an array of variables: input (*X*_1_—number of holes, *X*_2_—cutting speed, *X*_3_—feed, *X*_4_—time delay, *X*_5_—hole depth measuring point) and output (*Y*_1_—drilling temperature, *Y*_2_—hole diameter, *Y*_3_—out of roundness). The non-homogeneity of the variables was compensated by a linear normalization, in which all values of the variables were reduced to values in the range from 0 to 1, based on the following equations:(1)$$X_{i,j}^{norm}=\frac{X_{i,j}-X_{i.min}}{X_{i,max}-X_{i,min}},Y_{i,j}^{norm}=\frac{Y_{i,j}-Y_{i,min}}{Y_{i,max}-Y_{i,min}}$$where $X_{i,j}^{norm},X_{i,j},X_{i,max},X_{min,i},Y_{i,j}^{norm},Y_{i,j},Y_{i,max},Y_{i,min}$ are the normalized and measured maximum and minimum values of the input (*X*) and target (*Y*) variables, respectively.

The training stage was based on the dataset consisting of 4752 measured results (Table 5). The dataset was divided into subsets from 20 to 1000 data points, depending on the number of neural networks in the training. The overall measured data were applied as follows:-From 50 to 70% to train new neural networks;-From 15 to 25% to control overfitting of neural networks;-From 15 to 25% to test trained neural networks.

The normalized results of the measurements were included in the subsets on a random basis. In cases when the number of neural networks undergoing the training was more than 50, the bootstrap method was used to form the subsets [88].

To select a method for training the neural networks for MLP networks, conjugate descent, gradient descent, and BFGS (Broyden–Fletcher–Goldfarb–Shanno) were examined; for RBF networks, two-stage optimization algorithms were analyzed. At the beginning of the training, to ensure that a network behaves similar to a linear model, the maximum values of the weights of the synapses were restricted and initialized by uniformly distributed values in the range from 0 to 1. The final values of the weights were determined based on the training results.

When generating new neural networks of the MLP type, activation functions were varied. In different combinations for the hidden and the output layers, the following functions were selected: identity, logistic sigmoid, hyperbolic tangent, exponential, sine. For networks of the RBF type, activation functions did not change, and the Gaussian function was always chosen for the hidden neurons and a linear function for the output neurons.

The training was aimed to minimize the sum-of-squares error:(2)$$E=\overset{N}{\underset{1}{\sum }}{\left(Y_{i,j}^{norm}-Y_{i,j}^{targ}\right)}^{2}+\Delta_{W}\;$$where $Y_{i,j}^{norm},Y_{i,j}^{targ}$ are normalized and predicted values of variables, respectively, for the entire training set of N elements; $\Delta_{w}$ is an adjustment that increases the error for synapses with a heavier weight and is aimed at eliminating network overfitting,
(3)$$\Delta_{w}=w\times a/2\;$$
where *w* is the weight of the synapse; a is the constant for excluding weights, and it was set 0.001 and 0.0001 for the hidden and output layers, respectively.

## 3. Experimental Results and Discussion

### 3.1. Drilling Temperature

In this section, the drilling temperature of the CFRP/Ti alloy stack is investigated according to the measuring methodology described in Section 2.3. Additionally, the analysis of drilling temperature under different cutting conditions according to the DOE is discussed.

The dependencies of drilling temperature on the drilling conditions are shown in Figure 5. While the drilling temperature appeared highly dependent on drilling depth and time of delay, the cutting speed and feed rate appeared relatively affectless.

The temperature of the titanium layer drilled under (*v* = 15 m/min; *f* = 0.02 mm/r) raised up to 341.8 °C. In the tests № 6 and № 8, the drilling temperatures reached 461.8 °C and 477.3 °C, respectively. In the tests № 2, 3, 4, 5, 7, 9 the drilling was provided under HDM condition. In [11], the reduction in drilling temperature was reported when machining the CFRP layer after the Ti layer. With respect to the cutting condition, the drilling temperature was reduced from 280.2–265.5 °C under (*v* = 65 m/min; *f* = 0.08 mm/r) to 199.9–154.5 °C under (*v* = 15 m/min; *f* = 0.08 mm/r) with deviation of 45.4 °C. Difference in the speed of heat transfer in test № 9 (*v* = 65 m/min; *f* = 0.08 mm/r) and in test № 3 (*v* = 15 m/min; *f* = 0.08 mm/r) is explained by variation in machining time of CFRP layer, which for test № 9 was 2.2 s and for test № 3 was 9.45 s. A similar drilling temperature reduction in the CFRP layer was reported in [10,40,46]. This proves that the drilling temperature is directly proportional to the cutting speed due to the shortened duration of heat generation.

The temperature at the hole entrance varied from 50 to 200 °C, while it was from 350 to 600 °C at the hole exit. It is worth noting that the temperature keeps nearly constant within the CFRP domain but rises sharply after passing the adhesive interface with the Ti plate. The significant heat accumulation during the drilling of titanium raises concerns about joint integrity. It can be seen in Figure 5b that the temperature at the CFRP/Ti interface exceeds 300 °C, which is enough for the thermal degradation of adhesive agents. This may lead to defective holes with weakened interfacial adhesion. In summary, the drilling depth is the factor that strongly affects the CFRP/Ti drilling temperature.

The reduction in drilling temperature and heat distributions were observed when drilling holes № 23 under the test conditions (Figure 6). However, because of the drill bit wear [21,27], the values of the drilling temperature increased in all tests. When drilling CFRP/Ti alloy stack under CDM technique, the maximum drilling temperature in hole № 23 in CFRP was 121 °C, while for Ti alloy layer it reached up to 514 °C were measured in test № 8 (*v* = 65 m/min; *f* = 0.05 mm/r). The maximal drilling temperature under HDM conditions was measured in test № 7 (*v* = 65 m/min; *f* = 0.02 mm/r), for CFRP it was 313 °C, and for Ti alloy it was 699.4 °C. The maximal increments of drilling temperature within CFRP and Ti alloy were 18% and 11%, respectively, obtained under CDM conditions with cutting parameters (*v* = 15 m/min; *f* = 0.02 mm/r). The increment can be explained by the relatively long cutting length because of the combination of low cutting speed and feed. In tests under HDM conditions, the drilling temperature increment for CFRP compared to results for hole № 4 rose from 8% (11 °C) to 17% (46 °C). In tests № 3 and № 9, the effect of negative drilling temperature increment was observed. In such a way, the drilling temperature decreased up to 4% in these tests, which can be explained by more favorable conditions for heat transfer in CFRP under higher feed [40].

There is a remarkable reduction in drilling-induced temperature at each point within the CFRP/Ti stack as the time of delay becomes longer (Figure 7). The lowest temperature was observed in tests 1, 6, and 8 (<100 °C), where the time of delay was the longest (120 s). The longer delay ensured drill bit cooling and allowed the drilling process with the least thermal impact. On the other hand, the 5 s and 10 s delay resulted in nearly three times higher temperature as compared to the 120 s delay. The effect of the drill bit temperature was observed at both hole’s entrance and exit. Evidently, the time of delay prior to the drilling of the next hole is one of the key process parameters to control the temperature.

In tests conducted under the HDM conditions, the trend to increase drilling temperature dispersion in CFRP and Ti alloy was observed. The difference between the maximum and minimum drilling temperature under dispersion was considered. The drilling temperature dispersion ranged from 136 °C to 325 °C in CFRP and from 228 °C to 518 °C in Ti alloy. The effect of drilling temperature dispersion in CFRP can be explained by high drill bit temperature after drilling of Ti alloy layer in the previous hole. A similar phenomenon was reported in studies [10,40]. It can be concluded that HDM may cause thermal degradation in CFRP when the drilling temperature exceeds the glass transition temperature of the epoxy resin *T_g_* ≈ 180 °C. The effects of the cutting speed and feed rate on the temperature were negligible as compared to the drilling depth and delay time. Although a fractional DoE was used, it is evident in Figure 5 (e.g., compare the test 3 vs. 7) that the combined effect of the elevated cutting speed and feed elevates the temperature for no more than 20% at any hole depth. To recap, cutting speed and feed rate have an inferior influence on drilling temperature compared to the drilling depth and time delay.

Finally, temperature measurements in the present study were compared to those previously reported by other researchers and found similar. Xu and El Mansori [40] reached 100 °C under *v* = 20 m/min; *f* = 0.015 mm/r corresponds to the temperature and conditions we used in test 1. Zitoune [38] drilled under *v* = 38 m/min; *f* = 0.1 mm/r, which provides the same machining time as in tests 6 and 8, and thus similar conditions for heat generation. Their measured temperature results coincide with our results in Figure 5. Shao et al. [56] reported that the temperature of CFRP under *v* = 23 m/min; *f* = 0.005 mm/r was around 180 °C, which can be found on our temperature map for the drilling conditions. Overall, the measured results reported here coincide with the previously reported, which confirms the reliability of the extracted dependencies.

### 3.2. Hole Accuracy

The results of hole diameter measurements in CFRP and Ti alloy layers are shown in Figure 8. Concerning cutting parameters, the hole diameter ranged from 10.025 to 12.344 mm in CFRP and from 10.001 to 10.144 mm in Ti alloy. An et al. [10] described the influence of Ti alloy chip sliding on the oversizing of a hole in CFRP when drilling CFRP →Ti sequence. An increase in the feed caused an increase in hole diameter in the CFRP layer [13,22]. Higher feed ensures thicker chip to slide by CFRP hole wall [13]. The authors in [9,89] pointed out that the drilling sequence CFRP →Ti or Ti → CFRP is a significant factor that affects hole diameter because of the chip removal mechanism. However, the distribution of hole diameter in Ti alloy was not sensitive to feed. Simultaneously, the analysis of the trends of drilling temperature (Figure 7) and hole diameter in Ti alloy (Figure 8b) indicates that higher drilling temperature resulted in larger hole diameter. This is presumably due to the thermal expansion of the drill bit [11]. The effect of heat diffusion on hole diameter was reported by Wang et al. [9]. An et al. [10] and Dahnel et al. [42] reported the effect of tool wear on hole diameter, but it should be considered that the tool wear elevates the drilling temperature as well. These phenomena are more severe in the dry machining process, that is used in the aeronautic industry [90].

It was found that the time delay factor significantly influences hole out of roundness in the Ti alloy layer (Figure 9). HDM with 5 s and 10 s of time delay resulted in elevated out of roundness of holes in Ti alloy due to the thermal expansion of the drill bit, as explained in [11]. Importantly, we observed that the out of roundness in Ti was larger as compared to that in CFRP. This observation is similar to those reported in [8,22,42] and contradictive to those in [9,10,21], where the drill bit temperature was not taken into account. Herein, we show that these controversial results are due to the different times of delay.

The feature analysis of hole out of roundness measurements underlined that the distribution of values in Ti alloy increased under HDM conditions (Figure 10b). The maximum distribution of 0.240 mm was observed in test № 7 that corresponded to the maximum drilling temperature. In out of roundness of holes in Ti alloy was proportional to the cutting speed. Under CDM conditions, the maximal distribution of 0.261 mm was measured in test № 1, owning to the relatively high machining time and a significant effect of chisel edge allocation and radial run out of the drill bit. However, increasing the cutting speed up to 65 m/min in test № 8 led to a 0.110 mm distribution of out of roundness, which corresponded to high drilling temperature in Ti alloy (Figure 7). The analysis of measurements in CFRP shows that the HDM conditions result in a higher hole out of roundness. Under CDM conditions, the distributions of value were 0.084 mm, 0.153 mm, and 0.109 mm in tests № 1, 6, 8, respectively (Figure 10a). Though, under HDM conditions, the out of roundness was 0.884, 0.475, 0.121, 0.236, 0.100, and 0.723 mm in the tests, respectively (Figure 10a). The increased out of roundness can be explained by the thermal softening of CFRP [55] due to the heat transfer from the preheated drill bit and drilling-induced temperature. The thermal softening of CFRP is accompanied by the decrease in interfacial shear strength and fracture toughness which make it sensitive to sliding Ti alloy chips [43]. Higher distributions of out of roundness corresponded to relatively low cutting speed, which extended the time of Ti alloy chip contact with CFRP hole wall.

### 3.3. ANN Analysis of Experimental Results

The training was carried out in several iterative cycles (epochs). The maximum value of the cycles was taken as equal to 10,000 and stayed the same in all the calculations. With each iteration cycle, the synapse weights were selected in such a way as to reduce the learning error. If during 20 training cycles the error E, determined from the training sample, decreased by less than 10-7, then the training ended. In order to eliminate the overfitting of a network, the learning process also stopped in cases where an analogous error, calculated for the control sample where a = 0, was observed to increase (even if the error E continued to decrease).

The neural networks training results are shown in Table 6 and are illustrated in Figure 11. The trained neural networks were ranked according to the mean prediction error calculated for the test sample at a = 0. Finally, the network with the MLP 5-10-3 architecture, which for the test sample had the smallest mean error ¯E = 0.00388 and the correlation coefficients between the output and predicted values of variables that were close to one (r1 = 0.955 for variable *Y*_1_, r2 = 0.923 for variable *Y*_2_, and r3 = 0.725 for variable *Y*_3_), was chosen.

Feature ANN analysis of experimental results for one response value, namely drilling temperature (*Y*_1_), hole diameter (*Y*_2_), and hole out of roundness (*Y*_3_), was focused on the definition of each output (response) value sensitivity to factors *X*_1_–*X*_5_ (Table 5) verified in the present study. The most precise architecture of ANN was defined as MLP 5-10-3 with hyperbolic tangent function, which activates hidden neurons, exponential function for activation of output neurons, and gradient learning algorithm. This architecture was used for feature analysis of the sensitivity of drilling temperature, hole diameter, and out of roundness to factors, namely cutting speed, feed, time delay, and also hole number and hole depth. However, hole number and hole depth were not classical factors considered in the design of the experiment. The sensitivity to the number of holes can be considered as a tool wear effect, while the depth of the hole corresponds to the stack layer sequence. The sensitivity of response values was evaluated in relative units, where “1” means that response is not sensitive to factors. The scale of sensitivity of analyzed responses corresponds to the correlation coefficient of the control subset. With the decreasing correlation coefficient of the control subset, the sensitivity scale was also reduced (Figure 12a).

Based on the ANN analysis (Figure 12b), it was defined that drilling temperature is sensitive to hole depth (14.37 points). Such effect is explained by the fact that drilling temperature significantly differs when drilling CFRP and Ti alloy layers. The next factor to which drilling temperature was sensitive was cutting speed (4.33 points). The nature of cutting speed influence on drilling temperature was discussed earlier in chapter 3.1 of the present article. The last factor that affected the drilling temperature was time delay (3.56 points). In such a way, it can be concluded that the time period between adjacent holes is an essential factor that should be considered in feature studies focused on drilling CFRP/Ti alloy. The value of hole diameter was mainly sensitive to feed (7.97 points). Such phenomenon was earlier reported by Tang et al. [43], An et al. [10], and Zhou L. et al. [12]. The sensitivity to hole depth corresponded to various physical and mechanical properties of CFRP and Ti alloy and was reported in [8,9,10,21,22,42], while the sensitivity of hole diameter to time delay was not earlier considered as the factor affecting the hole accuracy. Out of roundness was sensitive to the time delay factor (5.20 points). Such sensitivity is explained by the connection between time delay and drilling temperature. The HDM conditions with 5 to 10 s delay between drilling the adjacent holes was accompanied by heat accumulation in the drill bit and heat flow in CFRP. These conditions lead to a reduction in CFRP stiffness and hole wall in CFRP sensitivity to a sliding Ti alloy chip. The cutting speed and layer of stack material (hole depth) affected out of roundness equally (2.79 points of sensitivity).

## 4. Conclusions

Herein, the sensitivity of drilling temperature, hole diameter, and hole out of roundness to the cutting speed, feed rate, and time delay during drilling of the CFRP/Ti alloy stack by using ANN were studied. During the research, the drilling temperature was measured via a wireless device (WICUTEM) through a K-type thermocouple sensor embedded under the flank surface of the drill. Hole diameter and out of roundness were measured at CNC coordinate measuring machine. The sensitivity of the measured responses to the drilling parameters was calculated using ANN analysis based on the MLP networks via conjugate descent, gradient descent, and BFGS (Broyden–Fletcher–Goldfarb–Shanno) algorithms. The conclusions are as follows:The drilling temperature is significantly affected by the time delay factor during single-shot drilling of the CFRP/Ti alloy stack. Increasing the drilling delay time between the two adjusted holes from 5 s to 120 s decreased the maximum drilling temperature from 350 °C to 150 °C in CFRP and from 690 °C to 575 °C in the Ti plate. Thus, the drill bit temperature is the key factor in controlling the drilling temperature;The hole diameter and out of roundness are affected by the drilling temperature. When drilling with a short time delay and aggressive feed, the drilling-induced heat transfers into the CFRP/Ti workpiece, deteriorating its fracture toughness and chip formation mechanism, respectively, that eventually results in oversizing. The diameter and out of roundness in CFRP grown along with the feed rate from 10.064 to 12.344 mm, and from 0.142 to 0.894, respectively. Low drilling temperature ensures higher hole accuracy;The ANN analysis allowed quantifying the sensitivities of the hole quality parameters to the drilling parameters. The drilling temperature was sensitive to cutting speed (4.33 points) and time delay (3.59 points) when drilling CFRP →Ti sequence, with a correlation coefficient of 0.97. The hole diameter was affected mainly by feed (7.97 points) and time delay (3.73 points) with a correlation coefficient of 0.94, while hole out roundness changed under the influence of time delay (5.20 points) and cutting speed (2.79 points) with a correlation coefficient of 0.74. Thus, ANN has a great potential for optimizing cutting parameters and other operational conditions of drilling holes in CFRP/Ti alloy stacks in industrial production;The most accurate ANN architecture to accomplish such a task is MLP-5-10-3, with a hyperbolic tangent function to activate hidden neurons, an exponential function to activate output neurons, and a gradient learning algorithm that provides an error of prediction up to 0.00388.

Suggestion for future research: In future research, optimization of cutting parameters could be presented to provide researchers and engineers with specific machining parameters for one short drilling strategy for machining holes in CFRP/Ti alloy layers. The optimization will focus on defining optimal values of cutting speed, feed rate, and time delay to reduce thermal effects on hole quality parameters.

## Figures and Tables

**Figure 1 materials-15-01940-f001:**
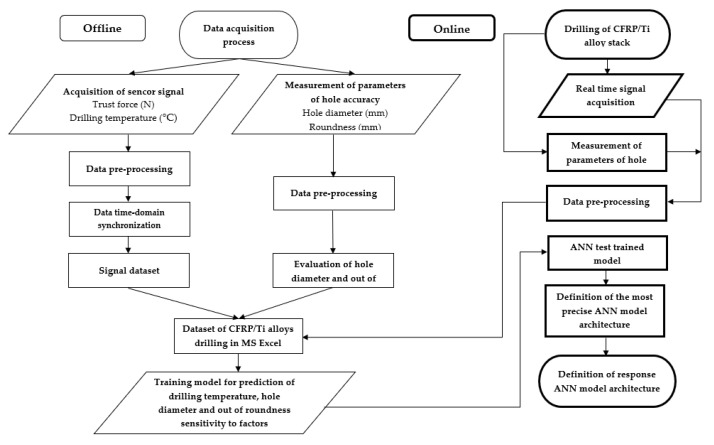
The procedure of experimental data acquisition and processing.

**Figure 2 materials-15-01940-f002:**
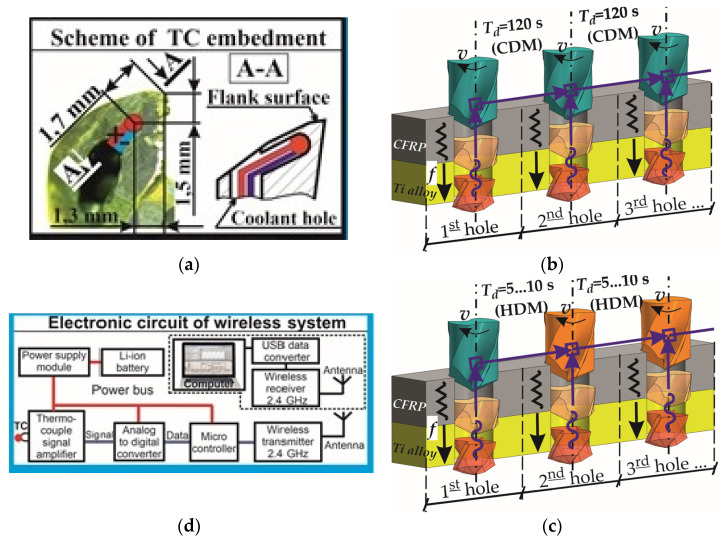
Principal scheme of drilling of CFRP/Ti alloy stack: (**a**) scheme of TC embedment; (**b**) technological scheme of drilling under “cool drill machining” condition (CDM); (**c**) technological scheme of drilling under “hot drill machining” condition (HDM); (**d**) scheme of electronic circuit of wireless system.

**Figure 3 materials-15-01940-f003:**
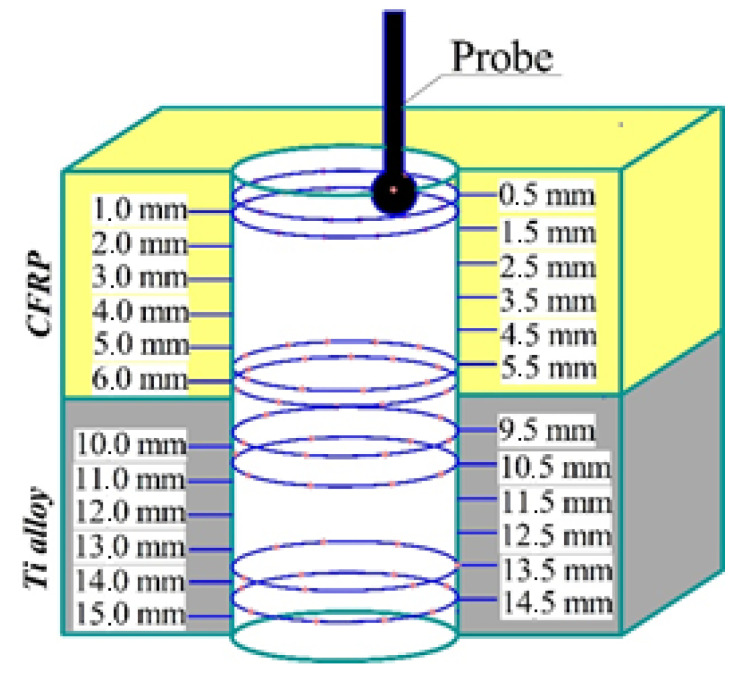
Measurement scheme of hole diameter.

**Figure 4 materials-15-01940-f004:**
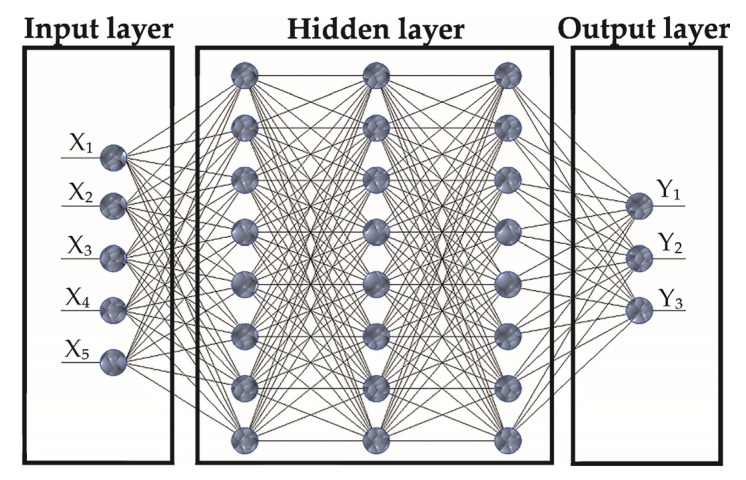
The architecture of the neural network.

**Figure 5 materials-15-01940-f005:**
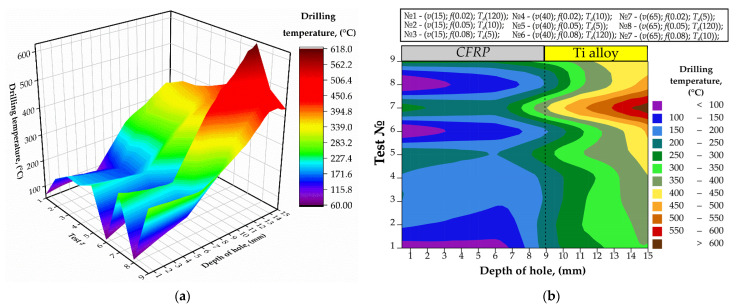
The drilling temperature measured in the tests № 1–№ 9 for the hole № 4: (**a**) 3D graph; (**b**) 2D graph.

**Figure 6 materials-15-01940-f006:**
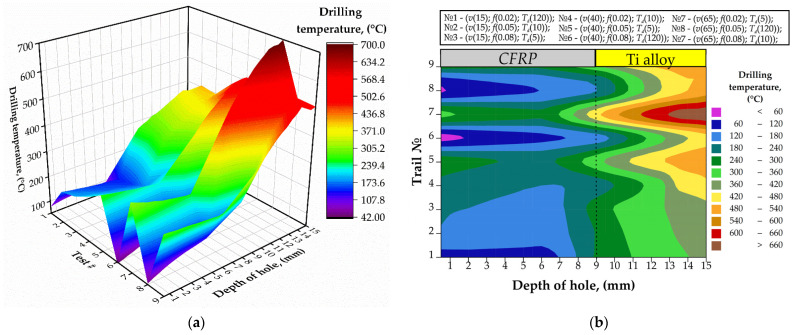
The drilling temperature measured in the tests № 1–№ 9 for the hole № 23: (**a**) 3D graph; (**b**) 2D graph.

**Figure 7 materials-15-01940-f007:**
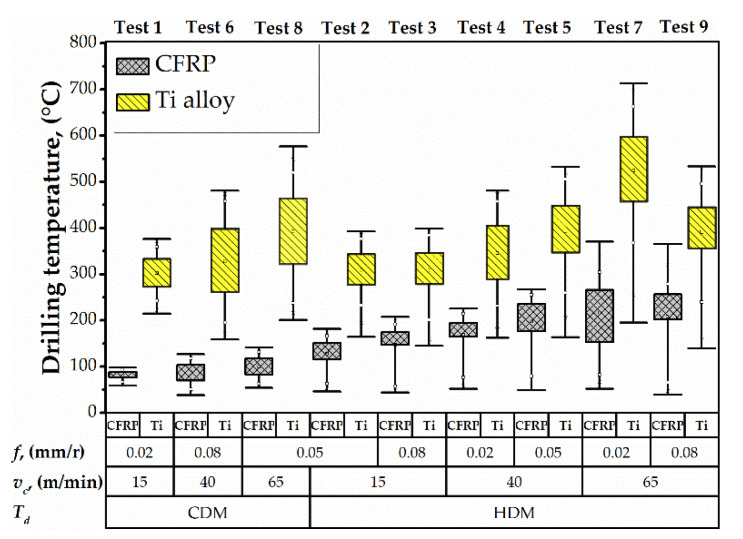
Dispersion of drilling temperatures.

**Figure 8 materials-15-01940-f008:**
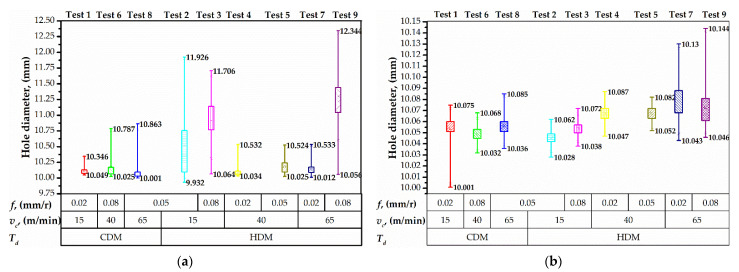
Hole diameter distribution in CFRP (**a**) and Ti alloy (**b**).

**Figure 9 materials-15-01940-f009:**
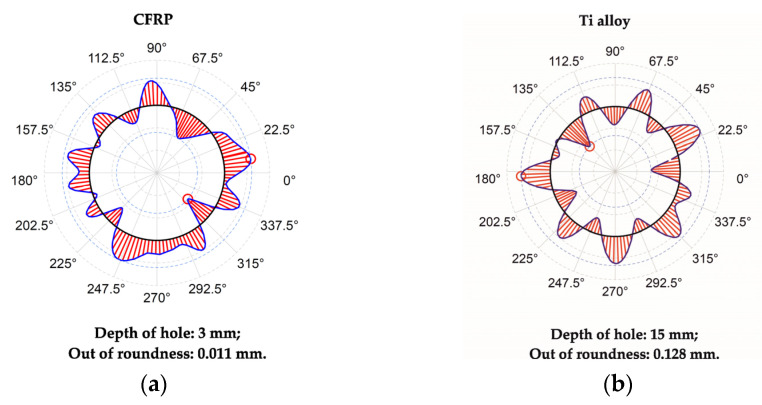
Section of out of roundness: (**a**) in CFRP; (**b**) and in Ti alloy in test № 8 hole № 9.

**Figure 10 materials-15-01940-f010:**
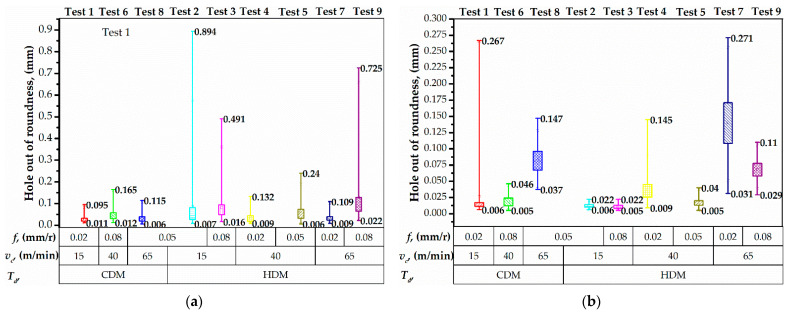
Out of roundness distribution in CFRP (**a**) and Ti alloy (**b**) layers.

**Figure 11 materials-15-01940-f011:**
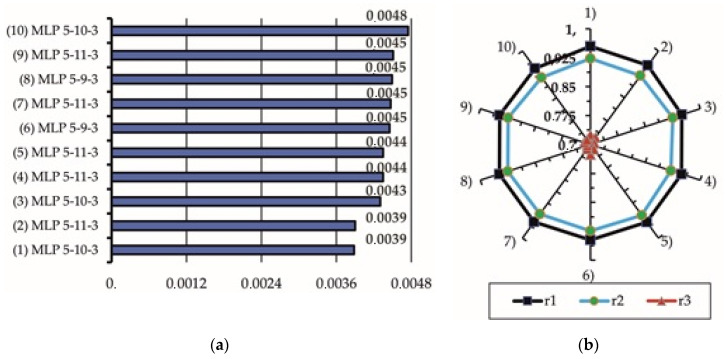
Average prediction errors for the test samples (**a**) and correlation coefficients r1, r2, and r3 in-between the output and predicted values (**b**).

**Figure 12 materials-15-01940-f012:**
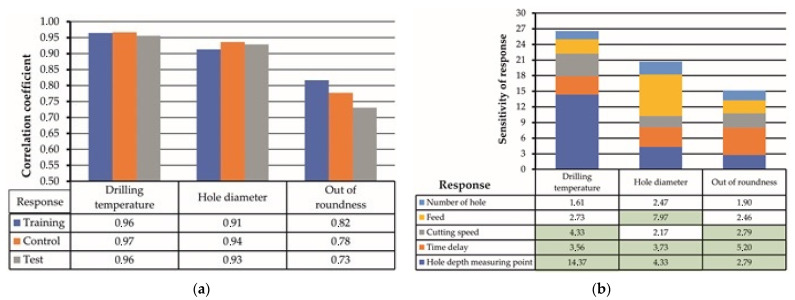
Correlation coefficient of output data (response) (**a**), and evaluation of response values sensitivity to factors (**b**).

**Table 1 materials-15-01940-t001:** Chemical compositions of Ti-2.5Al-2Mn alloy.

Ti	Al	C	O	Si	Mn	Fe
96.42%	1.92%	0.21%	0.19%	0.17%,	0.89%	0.20%

**Table 2 materials-15-01940-t002:** Drill bit geometry.

Geometric Parameters	Drill Number in Respect to the Test Number								
	1	2	3	4	5	6	7	8	9
*D*, (mm)	10.008	10.003	10.003	10.003	10.000	10.000	10.000	10.000	10.000
Radial runout, (mm)	0.010	0.012	0.016	0.008	0.008	0.008	0.008	0.008	0.008
Point angle, (*θ*°)	140.52	140.61	140.35	140.30	140.84	140.60	140.60	140.60	140.60
Axial relief angle, (*α*_a.r_°)	7.54	7.42	8.18	20.14	7.58	7.54	8.26	8.52	7.50
Chisel edge angle, (*ψ*°)	44.21	45.33	55.17	24.62	55.62	54.40	52.27	53.59	58.33
Helix angle, (*ω*°)	30.00	30.09	29.92	29.93	29.91	29.99	29.81	29.97	30.06

**Table 3 materials-15-01940-t003:** Design of experiment according to Taguchi orthogonal array L_9_ [11].

Drilling Performance (Factors)		Levels of Factors		
		1	2	3
A	Cutting speed, *v_c_* (m/min)	15	40	65
B	Feed, *f* (mm/r)	0.02	0.05	0.08
C	Time delay, *T_d_* (s)	120	10	5

**Table 4 materials-15-01940-t004:** Decoding of design of experiment (DOE).

Test №	Cutting Speed, *v* (m/min)	Feed, *f* (mm/r)	Time Delay, *T_d_* (s)	Remark to Time Delay
1	15	0.02	120	CDM
2	15	0.05	10	HDM
3	15	0.08	5	HDM
4	40	0.02	10	HDM
5	40	0.05	5	HDM
6	40	0.08	120	CDM
7	65	0.02	5	HDM
8	65	0.05	120	CDM
9	65	0.08	10	HDM

**Table 5 materials-15-01940-t005:** Dataset of CFRP/Ti alloy stack drilling in MS Excel (fragment).

	A	B	C	D	E	F	G	H	I
1	**Test №**	**Hole №**	**Cutting Speed, *v* (m/min)**	**Feed, *f* (mm/r)**	**Time Delay, *Tt* (s)**	**Hole Depth Measuring Point, (mm)**	**Drilling Temperature, *t* (°C)**	**Hole Diameter,** ***D* (mm)**	**Hole Out of Roundness, Δ*D* (mm)**
2		** *X* _1_ **	** *X* _2_ **	** *X* _3_ **	** *X* _4_ **	** *X* _5_ **	** *Y* _1_ **	** *Y* _2_ **	** *Y* _3_ **
3	1	1	15	0.02	120	0.5	58.68	10.118	0.031
4	1	1	15	0.02	120	1	61.26	10.139	0.022
5	1	1	15	0.02	120	1.5	63.37	10.178	0.026
6	1	1	15	0.02	120	2	65.06	10.184	0.031
7	1	1	15	0.02	120	2.5	67	10.202	0.053
8	1	1	15	0.02	120	3	67.44	10.215	0.062
9	1	1	15	0.02	120	3.5	67.68	10.239	0.067
10	1	1	15	0.02	120	4	67.79	10.282	0.082
4745	9	23	65	0.08	10	10.5	422.14	10.097	0.048
4746	9	23	65	0.08	10	11	442.94	10.079	0.052
4747	9	23	65	0.08	10	11.5	460.33	10.064	0.055
4748	9	23	65	0.08	10	12	474.01	10.065	0.055
4749	9	23	65	0.08	10	12.5	494.37	10.063	0.063
4750	9	23	65	0.08	10	13	508.78	10.065	0.061
4751	9	23	65	0.08	10	13.5	529.73	10.057	0.063
4752	9	23	65	0.08	10	14	533.29	10.055	0.065
4753	9	23	65	0.08	10	14.5	504.25	10.056	0.07
4754	9	23	65	0.08	10	15	510.09	10.057	0.077

**Table 6 materials-15-01940-t006:** Learning outcomes of the top ten neural networks.

№	Architecture	Function to Activate Hidden Neurons	Function to Activate Output Neurons	Learning Algorithm	Sensitivity				
					*X* _1_	*X* _2_	*X* _3_	*X* _4_	*X* _5_
1	MLP 5-10-3	Hyperbolic tangent	Exponential	Gradient	1.422	3.534	2.276	4.802	16.83
2	MLP 5-11-3	Logistic sigmoid	Logistic sigmoid	BFGS	1.979	7.209	2.74	8.326	14.37
3	MLP 5-10-3	Logistic sigmoid	Exponential	BFGS	1.663	3.164	1.694	46.87	12.62
4	MLP 5-11-3	Hyperbolic tangent	Exponential	BFGS	1.239	2.445	1.195	6.851	14.32
5	MLP 5-11-3	Identity	Logistic sigmoid	BFGS	1.243	2.941	1.401	29.51	13.15
6	MLP 5-9-3	Logistic sigmoid	Exponential	BFGS	2.223	1.965	1.254	7.033	14.04
7	MLP 5-11-3	Hyperbolic tangent	Logistic sigmoid	BFGS	1.521	4.18	1.792	4.207	12.28
8	MLP 5-9-3	Logistic sigmoid	Exponential	Gradient	1.225	2.845	1.859	3.878	12.52
9	MLP 5-11-3	Logistic sigmoid	Logistic sigmoid	Gradient	2.123	3.669	1.771	3.469	12.27
10	MLP 5-10-3	Exponential	Logistic sigmoid	BFGS	1.803	3.373	2.847	13.75	10.63

## Data Availability

Not applicable.

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
