# Peer review of "Application of ANN for Analysis of Hole Accuracy and Drilling Temperature When Drilling CFRP/Ti Alloy Stacks"

_materials, 2022, doi:10.3390/ma15051940_

Round 1
Reviewer 1 Report
The manuscripts discuss the drilling of CFRP/Ti (Carbon Fiber-Reinforced Plastic/ Titanium alloy). It is true that poor machinability of stack materials in combination with the inhomogeneous behavior of CFRP and Ti alloy face manufacturing problems. The following comments have been made in order to elevate the quality and readability of this manuscript.
- Make sure that the research gap and problem have been identified properly in the introduction section. Thus, it should clearly reflect after your literature reviews.
- The last passage of the introduction section should be elaborated in order to state the benefits of ANN applications.
- What type of CNC machine was used? Please add some details of the machine details.
- Did you consider uncertainties in data acquisition during drilling and temperature recording operations?
- Please mention the thermocouple installation and temperature recording process.
- Explain the purpose of choosing MLP/RBF ANN architecture.
- The contour legends and resolution of all figure needs revision. The legends should appear clearly.
- In the last point of conclusion: could you please discuss the importance and significance of the study?
- Please improve the language and correct any grammar mistakes in the manuscript.
Author Response
Dear Opponent,
Thank you for your comments. We have also compiled our responses and comments into a file - attached.

Reviewer 2 Report
The level of English language in this paper is insufficient for me to review it.
Author Response

(The authors gave the same response as above.)

Reviewer 3 Report
In the present work, the influence of the process parameters on the temperature and the hole accuracy during drilling has been analysed. In particular, the authors produced some Ti/CFRP stacks and then drilled them considering different process parameters, such as cutting speed, feed rate and time delay between two consecutive holes. The temperature reached during drilling was recorded by a thermocouple embedded in the drill, the thrust force was evaluated by a dynamometer and the hole geometry was evaluated by using a CMM, after machining. All these data were analysed through ANN. The topic of the work is interesting since drilling of composite and hybrid materials is needed for the realization of assemblies. Moreover, the description of the performed experimental analysis appears quite exhaustive. Due to these reasons, the referee is positive toward this submission, nevertheless, in the referee’s opinion there is still room for some improvements, and the following revisions are warmly encouraged:
- The thrust force has been measured but the relevant results have not been commented.
- Drilling temperature has been investigated in several works, such as “Sorrentino, L., Turchetta, S., Colella, L., Bellini, C. Analysis of Thermal Damage in FRP Drilling (2016) Procedia Engineering, 167, pp. 206-215. DOI: 10.1016/j.proeng.2016.11.689” and “Bolar, G., Sridhar, A.K., Ranjan, A. Drilling and helical milling for hole making in multi-material carbon reinforced aluminum laminates (2022) International Journal of Lightweight Materials and Manufacture, 5 (1), pp. 113-125. DOI: 10.1016/j.ijlmm.2021.11.004”
Author Response

(The authors gave the same response as above.)

Reviewer 4 Report
The drilling machining of CFRP/Ti alloy stacks was studied and described in the submitted manuscript. My comment is as follow.
Many experiments were done in this study and the influences of cutting speed, feed speed and time delay on cutting performance were explained. Could you give the optimized cutting parameters to make the best cutting performance or give idea to make it?
Author Response

(The authors gave the same response as above.)

Reviewer 5 Report
I am asking the authors to explain what assumptions were made for the numerical analysis of the drilling process.The paper presents data on the verification of the results of numerical analysis and the actual drilling process, however, the error value of calculations made in accordance with the presented method is not given.
I also ask the authors for information on how long it takes to perform the numerical analysis with the presented method.In the conclusions, it is advisable to provide recommendations for the selection of drilling parameters so as to ensure the best accuracy and quality of the hole.In the opinion of the Reviewer, it will be interesting to continue work in the direction of research indicated by the Authors.
Author Response
Response to Reviewer 5 Comments (28.02.2022)
Point 1. I am asking the authors to explain what assumptions were made for the numerical analysis of the drilling process. The paper presents data on the verification of the results of numerical analysis and the actual drilling process, however, the error value of calculations made in accordance with the presented method is not given.
Response 1: Response 1: Thank you for your comment. Here is our explanation. The prediction/calculation errors were specified in Figure 11. Based on the smallest mean error E= 0.00388 (paragraph under Figure 11) the best ANN architecture was chosen.
Point 2. I also ask the authors for information on how long it takes to perform the numerical analysis with the presented method.In the conclusions, it is advisable to provide recommendations for the selection of drilling parameters so as to ensure the best accuracy and quality of the hole.In the opinion of the Reviewer, it will be interesting to continue work in the direction of research indicated by the Authors.
Response 2: Thank you for this comment. We give our explanation. One of the goals of the current article was to define the sensitivity of drilling temperature, hole diameter and out of roundness to factors: cutting speed, feed, time delay. This was proposed to do via ANN. The results of ANN experimental data processing satisfied the authors. The authors didn’t count how many minutes or hours ANN analysis was calculated. In such a way author couldn’t specify such information in the paper. The recommendations concerning recommended drilling parameters can not be mentioned in this paper because it is necessary to provide optimization of experimental results. So far the optimization desires to be an objective for one more article these results will be reported in future work. From the data obtained so far on the effect of temperature on the accuracy of the drilled hole, we discuss this in the conclusion. It is recommended to choose cutting parameters that ensure the generation of the lowest possible cutting temperature ( conclusion points 2 and 3). A low cutting temperature and thus a more accurate hole is achieved by low cutting speeds and small feed rates velocity.
Reviewer 6 Report
Paper can be accepted after following corrections:
- Figures 1 and 2 should be presented accordingly to the technical standards. Please re-draw. Small drawing of measuring elements should be removed, whereas information flow should be clearly indicated.
- Figure 3 is not suitable for the scientific publication. Please remove. However, measuring scheme should be clearly explained in the text.
- Please explain the criteria for choose of the architecture of ANN
- figures 6a and 6b seems to be doubled. Please remove or clearly explain.
Author Response
Point 1. Figures 1 and 2 should be presented accordingly to the technical standards. Please re-draw. Small drawing of measuring elements should be removed, whereas information flow should be clearly indicated.
Response 1: We thank the respondent for pointing out the alignment of Figure 1 and Figure 2 with the technical standards. We have used normalized elements for drawing these diagrams. Figure 1 has been redrawn so that the flow of information is clear and the small images of the measurement elements have been removed and the text about the measured elements has been retained. Figure 2 has been amended in light of the objector's comments. Only the principle diagrams remain and the textual commentary has been modified.
Point 2. Figure 3 is not suitable for scientific publication. Please remove. However, the measuring scheme should be clearly explained in the text.
Response 2: Yes, one can agree with the opponent that images showing, for example, a machine tool, etc. are not suitable for scientific publication. Therefore, we have removed the photographs of the measuring equipment from Figure 3 and left the diagram of the hole diameter measurement in the figure, which is also explained in the text.
Point 3. Please explain the criteria for choose of the architecture of ANN
Response 3: Thank you for your comment. Here is our explanation. The criteria for choosing the most accurate ANN architecture was mean square error E. The most accurate architecture has E=0.00388, which was mentioned in the text under figure 11.
Point 4. Figures 6a and 6b seems to be doubled. Please remove or clearly explain.
Response 4: Thank you for your comment. We have carefully reviewed your comment and here is our explanation. Figure 6b is a 2D projection of Figure 6a. The authors believe that the 3D representation of the graph in Figure 6a makes it easier to understand what is shown in Figure 6b. We feel that both Figures 6a and 6b complement each other appropriately and give a wider range of possibilities for evaluating the data obtained. The opponent mentions the duplication of the figures. This merging of the figures could relate more to Figures 5 and 6 with each other. Thus, at first sight, it may indeed appear so. Figure 5 and Figure 6 refer to a different sample each time. On closer examination, it is possible to see the differences between Figures 5 and 6, e.g. in the 3D and 2D plots, in the values of the drilling temperatures on their scales.
Round 2
Reviewer 2 Report
The revised version greatly improved in readability and the efforts from the authors are much appreciated.
The paper consists of two parts: an experimental study into the effects of drill parameters and an ANN analysis of the results. The authors fail to demonstrate the advantages of using ANN over any other (statistical) analysis technique. A comparison with traditional techniques such as ANOVA would be beneficial. The evaluation of various architectures does not add to the learnings of the article.
The conclusions from the experimental section are all straightforward in my view (e.g. temperature increases with feed rate and shorter time between drilling). Drilling depth is used as a parameter, while it simply relates to the material that is being drilled.
The huge variation in drill diameter (> 20 % !) is explained by thermal expansion of the drill bit but is probably more related to melting or thermal degradation of the CFRP material.
Why is correlation coefficient r3 so low?
Author Response
Response to Reviewer 2 Comments, Round 2
Point 1. The revised version greatly improved in readability and the efforts from the authors are much appreciated.
Response 1: We are pleased that we have met the necessary language requirements in this regard. We have made additional minor grammatical corrections (noted in the text with comments indicated). Thank you very much.
Point 2. The paper consists of two parts: an experimental study into the effects of drill parameters and an ANN analysis of the results. The authors fail to demonstrate the advantages of using ANN over any other (statistical) analysis technique. A comparison with traditional techniques such as ANOVA would be beneficial. The evaluation of various architectures does not add to the learnings of the article. The paper consists of two parts: an experimental study into the effects of drill parameters and an ANN analysis of the results. The authors fail to demonstrate the advantages of using ANN over any other (statistical) analysis technique. A comparison with traditional techniques such as ANOVA would be beneficial. The evaluation of various architectures does not add to the learnings of the article.
Response 2: Thank you for your comment! Yes, the opponent is correct that the paper is essentially in two parts. We just focused not on comparing statistical methods with each other but on the possibilities of using the ANN method. We think that we have been able to show that this method also has its justification to be used for the analysis of such processes. Yes, the comparison of several statistical methods also has its justification. Due to the scope of our paper, we did not go into the comparison of statistical methods (ANN, ANOVA, and others) but we thank our opponent for this comment, we think it is a very good topic for future research.
We would like to state that the choice of the method of artificial neural networks for the mathematical processing of experimental data is since it is modern, rapidly developing, and has recently been used to solve a wide range of applied problems. When choosing a method, the authors considered the works [1-5], which describe the positive results of using artificial neural networks to analyze data obtained in experiments similar to the experimental study. We admit that ANOVA can also be effective. However, the involvement of the mathematical apparatus of artificial neural networks made it possible to solve the problems posed in the article. Therefore, the feasibility of using other mathematical methods was not investigated. During ANN processing of experimental data, more than one hundred different ANN architectures were calculated. In the manuscript, ten most accurate architectures were reported to prove that authors made their conclusion based on the analysis of the most accurate one but not the only one ANN architecture.
References:
- Sathish Rao, U.; Raj Rodrigues, L.L. An application of dissimilar ann algorithms to improve the simulation performance of flank wear extrapolation in GFRP composite drilling. International Journal of Mechanical and Production Engineering Research and Development 2018, 8, 325-336, doi:10.24247/ijmperdoct201837.
- Parmar, J.G.; Dave, K.G.; Gohil, A.V.; Trivedi, H.S. Prediction of end milling process parameters using artificial neural network. In Proceedings of the Materials Today: Proceedings, 2020; pp. 3168-3176.
- Sada, S.O. Improving the predictive accuracy of artificial neural network (ANN) approach in a mild steel turning operation. International Journal of Advanced Manufacturing Technology 2021, 112, 2389-2398, doi:10.1007/s00170-020-06405-4.
- Wu, T.Y.; Lei, K.W. Prediction of surface roughness in milling process using vibration signal analysis and artificial neural network. International Journal of Advanced Manufacturing Technology 2019, 102, 305-314, doi:10.1007/s00170-018-3176-2.
- Cui, J.C.; Liu, W.; Zhang, Y.; Gao, C.Y.; Lu, Z.; Li, M.; Wang, F.J. A novel method for predicting delamination of carbon fiber reinforced plastic (CFRP) based on multi-sensor data. Mechanical Systems and Signal Processing 2021, 157, 23, doi:10.1016/j.ymssp.2021.107708.
Point 3. The conclusions from the experimental section are all straightforward in my view (e.g. temperature increases with feed rate and shorter time between drilling). Drilling depth is used as a parameter, while it simply relates to the material that is being drilled.
Response 3: Thank you for your comment! Yes, we agree with the opponent regarding the evaluation of the conclusions. The conclusions in the respective chapter of the manuscript summarized the study's results. They cannot be divided on findings of the experimental section and section of ANN analysis so far results of ANN analysis based on experimental results. However, conclusion #1 summarizes cutting parameters' effect on drilling temperature. The impact of the time delay factor underlined in conclusion #1 was not earlier reported in any publication (90 publications for the last five years were considered) in the field. The drilling depth was not considered a factor in the design of the experiment but was taken into account during ANN analysis. Considering drilling depth in ANN analysis was significant for processing measurements of drilling temperature each 0.5 mm for each of twenty-three holes in each of nine tests (4754 measurement results in total). Such detailed analysis of drilling temperature evaluation though the depth of hole makes it possible to compare results of drilling temperature and deviations of hole diameter and out of roundness in respective points of measurement (in the hole), and define the potential effect of drill bit thermal expansion. Similar results were not reported earlier.
Point 4. The huge variation in drill diameter (> 20 % !) is explained by thermal expansion of the drill bit but is probably more related to melting or thermal degradation of the CFRP material.
Response 4: Thank you for your comment. The thermal degradation of the CFRP was studied in experimental research through 3D tomography at METROTOM 1500. But these results have not been reported in the current manuscript yet and will be adequately discussed in future publications. For that reason, authors have no arguments justifiably assert that oversizing of hole diameter in CFRP is influenced by thermal degradation of epoxy yet. However, in raw 334-335, authors suspect that drilling temperature higher than 300°C could be enough for thermal degradation. This suspicion is based on earlier studies reported in the publications. At the same time, dependence between oversize of the hole in CFRP, law cutting speed, high feed, and law time delay was observed and discussed. Although the drilling temperature and thermal expansion of the drill bit depend on each other, the authors did not conclude that the oversizing of holes in CFRP is coursed by thermal expansion. The authors reported that increasing drilling temperature reduces CFRP's interfacial shear strength and fracture toughness, making it more sensitive to sliding Ti alloy chips (raw 426-431). In such a way, drill bit thermal expansion was not defined as significant for CFRP but affected the deviation of hole diameter in the Ti alloy layer.
Point 5. Why is correlation coefficient r3 so low?
Response 5: Thank you for a very good question. The low correlation coefficient of r3 (hole out of roundness) in comparison to r1 (drilling temperature) and r2 (hole diameter) is explained by the relatively low sensitivity of hole out of roundness to the factors examined in the present study. Such a conclusion was made based on ANN processing of experimental data.
